# ANCA-Associated Vasculitis: An Update

**DOI:** 10.3390/jcm10071446

**Published:** 2021-04-01

**Authors:** Salem Almaani, Lynn A. Fussner, Sergey Brodsky, Alexa S. Meara, David Jayne

**Affiliations:** 1Division of Nephrology, The Ohio State University Wexner Medical Center, Columbus, OH 43201, USA; 2Division of Pulmonary and Critical Care Medicine, The Ohio State University Wexner Medical Center, Columbus, OH 43201, USA; lynn.fussner@osumc.edu; 3Department of Pathology, The Ohio State University Wexner Medical Center, Columbus, OH 43201, USA; sergey.brodsky@osumc.edu; 4Division of Rheumatology, The Ohio State University Wexner Medical Center, Columbus, OH 43201, USA; alexa.meara@osumc.edu; 5Department of Medicine, University of Cambridge, Cambridg CB2 0QQ, UK; dj106@cam.ac.uk

**Keywords:** ANCA vasculitis, review, vasculitis, crescentic glomerulonephritis

## Abstract

Anti-neutrophil cytoplasmic antibody (ANCA)-associated vasculitis (AAV) represents a group of small vessel vasculitides characterized by granulomatous and neutrophilic tissue inflammation, often associated with the production of antibodies that target neutrophil antigens. The two major antigens targeted by ANCAs are leukocyte proteinase 3 (PR3) and myeloperoxidase (MPO). AAV can be classified into 3 categories based on patterns of clinical involvement: namely, granulomatosis with polyangiitis (GPA), microscopic polyangiitis (MPA) and eosinophilic GPA (EGPA). Clinically, AAV involves many organ systems including the lungs, kidneys, skin, and nervous system. The prognosis of AAV has improved dramatically due to advances in the understanding of its pathogenesis and treatment modalities. This review will highlight some of the recent updates in our understanding of the pathogenesis, clinical manifestations, and treatment options in patients with AAV focusing on kidney involvement.

## 1. Epidemiology

Anti-neutrophil cytoplasmic antibody (ANCA)-associated vasculitis (AAV) is a relatively rare disease with an estimated prevalence of 200–400 cases per million people [1,2,3,4]. The incidence of AAV has increased over time, likely due to improvements in ANCA testing, disease classification, and clinical recognition [3,5,6,7]. AAV affects males and females equally [1,2,3,4]. Patients with GPA (granulomatosis with polyangiitis) and MPA (microscopic polyangiitis), the most common clinical phenotypes of AAV, are typically older adults. In contrast, patients with EGPA (eosinophilic GPA) tend to be younger, perhaps related to its close association with asthma. AAV is less common in children [1,2,3,4,5,6,8].

GPA and MPA tend to follow a different geographical distribution. GPA is more common in patients with European descent, while MPA is more common in patients from Eastern Asia. Additionally, the incidence of GPA seems to increase with the increase in distance from the equator [2,4,9,10,11,12,13,14,15]. In multi-ethnic populations, AAV is more common in Caucasians [6,16,17].

## 2. Pathogenesis

The pathogenesis of AAV is complex and includes genetic, environmental, and infectious factors. Genome-wide association studies (GWAS) identified several genetic associations that include major histocompatibility complex (*MHC*) and non-MHC genetic loci. Those associations were better aligned with the ANCA subtype (leukocyte proteinase 3, PR3-ANCA versus myeloperoxidase, MPO-ANCA) than with the clinical syndrome (GPA versus MPA) [18,19]. For example, polymorphisms in the genes coding for HLA-DP, alpha-1 antitrypsin (*SERPINA1*, the major endogenous inhibitor of PR3), and PR3 (*PRTN3*) are associated with development of PR3-AAV in Europeans, while polymorphisms in the HLA-DQ gene was associated with MPO-AAV [18,19]. Different genetic loci were associated with AAV in other populations such as HLA-DRB1 in African Americans [17] and HLA-D55 in Asians [20]. In addition to genetic associations, several environmental exposures have been described in patients with AAV such as smoking [21], exposure to silica dust [22,23,24] and solvents [10]. Infection has long been thought to play a role in triggering disease activity. Nasal carriage of *Staphylococcus aureus* has been associated with a higher relapse rate in patients with GPA [25]. The mechanism by which *S. aureus* accentuates disease activity is not completely understood, however there is evidence to suggest that molecular mimicry might play a role as a peptide that is part of the plasmid-encoded 6-phosphogluconate dehydrogenase in *S. aureus* induces anti-MPO T cell immunity in mice [26]. Additional evidence for molecular mimicry has been described in patients with anti-lysosomal membrane protein-2 (LAMP-2) antibodies, which has complete homology with the bacterial adhesion molecule FimH [27].

The majority of patients with active AAV have circulating ANCAs [28], however ANCAs can be found in patients with other autoimmune and/or infectious conditions [29,30,31]. Nevertheless, there is ample evidence to support the pathogenicity of ANCAs. For example, transfer of anti-MPO IgG or splenocytes from MPO-immunized mice to *Rag2* deficient mice (lacking B and T-cells) results in crescentic glomerulonephritis (CGN) [32]. In addition, irradiation of *MPO* knockout mice after immunization with MPO followed by bone marrow transplantation also results in CGN [33]. Moreover, ANCAs facilitate glomerular leukocyte adhesion [33], neutrophil recruitment [34], MPO release [35], and induce the formation of neutrophil extracellular traps (NETosis) [36]. Additionally, ANCAs induce in vitro neutrophil degranulation [37], and stimulate neutrophils to damage human endothelial cells [38]. Clinical evidence supporting the pathogenicity of ANCAs is evident in patients with AAV, where ANCA target antigens (PR3, MPO, lactoferrin) can be localized within or around lesions of fibrinoid necrosis [39]. Additionally, a case of a newborn who developed pulmonary-renal syndrome after passive transfer of the mother’s MPO-ANCA was reported [40]. Moreover, therapies that deplete B-lymphocytes, discussed in more detail later, have been effective in treating AAV and further suggest that the antibodies may be pathogenic.

The production of antibodies targeting self-antigens and maintenance of autoreactive B-cells highlight the key role of loss of tolerance in AAV (Figure 1). How this occurs in AAV is incompletely understood, but multiple pathways have been proposed. Impaired clearance of apoptotic neutrophils may lead to prolonged exposure of autoantigens to circulating antigen-presenting cells [41,42,43]. Patients with AAV have lower levels of and/or dysfunctional B and T regulatory lymphocytes (Breg and Treg, respectively) [44,45,46,47]. Other stimuli, such as active infection and environmental exposures, may also lead to increased expression and release of MPO and PR3 as part of neutrophil extracellular traps (NETs), further exposing these antigens to antigen presenting cells [25,36,48]. Once ANCAs develop, survival of the autoantibody-secreting B-cells is dependent upon additional factors. For example, levels of the B-cell-activating factor (BAFF, also known as BLyS) are increased in AAV, which may support survival of autoreactive lymphocytes [49,50,51,52]. Additionally, genetic factors may contribute to the propensity for perpetuation of autoreactive lymphocytes. Polymorphisms in the major MHC genes have been associated with development of AAV, as have variants in PTPN22, which encodes the protein tyrosine phosphatase non-receptor type 22. PTPN22 affects the responsiveness of B and T cell receptors, and has been implicated in multiple other autoimmune diseases including lupus, rheumatoid arthritis, and type 1 diabetes [53,54,55,56].

In addition to B-cells and their autoantibodies, T-cells play a major role in the development of AAV. MPO-specific CD4+ T-cells induce CGN in mice [57], and depletion of CD4+ or CD8+ T-cells rescues mice from CGN despite the presence of ANCAs [58,59].

In addition, murine CGN can be induced even in the absence of ANCAs or B-cells [58], and Treg depletion accentuates kidney injury [60]. Th17 cells may also have a role in AAV and their responses seem to have a temporal role; inducing CGN early on but having a protective role in the later phase of injury by restraining Th1 responses [61]. In patients with AAV, those with a poor prognosis have a CD8+ T-cell expression profile that is enriched for T-cell and IL-7 receptor signaling [62]. Additionally, during active kidney vasculitis, the number of effector memory T-cells (CD4 + CCR7-) increases in urine and decreases in the blood, suggesting their migration from the blood to the kidney [63].

Monocytes and macrophages are also involved in the pathogenesis of AAV. Monocytes express MPO and PR3 and when engaged by ANCAs produce reactive oxygen species and proinflammatory cytokines [64,65]. CD68+ monocyte/macrophages infiltrate glomeruli and can produce extracellular traps [66]. Additionally, monocytes and macrophages are key components of granulomas [67]. Moreover, CD68 + CD163+ macrophages, which are involved in repair and fibrosis [68] are the most abundant macrophages at sites of segmental fibrinoid necrosis [69], suggesting a role in initiation of glomerular injury and fibrosis.

Once autoreactive lymphocytes and their antibodies engage their target antigens, a cascade of events that involve the alternative pathway of the complement system (ACP) ensues and facilitates development and further propagation of kidney injury. ANCA-activated neutrophils generate C5a, further propagating neutrophil activation and interaction with endothelial cell layers [70]. Inhibition of the ACP or the C5a receptor rescues mice from MPO-induced CGN [71]. Additionally, theC5a receptor blocker avacopan was efficacious in management of AAV (discussed later) [72,73]

## 3. Clinical Manifestations and Diagnosis

The hallmark of AAV is the presence of necrotizing inflammation and fibrinoid necrosis in the walls of small and medium-sized vessels, which may be accompanied by necrotizing granulomas in GPA, or eosinophilic infiltrates in EGPA. The clinical manifestations of AAV largely depend on the vascular bed affected, though the lungs and kidneys are the most common sites of involvement [74,75]. Involvement of the upper respiratory tract can cause symptoms of rhinitis, sinusitis, deafness or epistaxis. When the lower respiratory tract is involved, patients can suffer from alveolar hemorrhage that manifests as cough, hemoptysis, shortness of breath, and/or hypoxemia, typically with bilateral pulmonary infiltrates on imaging. In addition, some patients present with pulmonary nodules and/or cavities, or interstitial lung disease. Subglottic or bronchial stenosis may manifest with dyspnea, cough, or noisy breathing. Neurological involvement most commonly manifests as mononeuritis multiplex or hearing loss, though meningeal inflammation, cerebral vasculitis, and sensory peripheral neuropathies may also occur. The skin is often involved resulting in a skin rash, often with purpuric features. The kidneys are frequently involved (sometimes the sole organ involved—termed “renal-limited vasculitis”) resulting in hematuria, proteinuria, and an elevated serum creatinine level. Other less common manifestations include myocarditis and enteritis/colitis. Clinical manifestations of AAV have been used to classify AAV into 3 categories, which differ in the frequency of organs involved, predominant autoantibody, rate of relapse, and clinical outcome [76] (Table 1). In addition to organ-specific manifestation, patients with AAV often share non-specific features of systemic inflammation, such as fatigue, fever, weight loss, and arthralgias and myalgias. A prodromal phase of many months of constitutional disturbance often precedes clinical presentation, whereas in EGPA months or years of maturity-onset asthma and naso-sinusitis occur before vasculitic features.

Kidney involvement in AAV is common and can follow a slow or rapidly progressive course, in some cases necessitating initiation of dialysis. A kidney biopsy is preferred when possible to confirm the diagnosis. Histological exam of kidney tissue with light microscopy typically reveals necrotizing crescentic glomerulonephritis (Figure 2), which usually involves some but not all of the glomeruli. Uninvolved glomeruli usually appear normal without endocapillary or mesangial hypercellularity. The tubulointerstitium often demonstrates an inflammatory cellular infiltrate. Indirect immunofluorescence reveals little or no staining for immunoglobulins or complement, hence the descriptive term “pauci-immune” glomerulonephritis. 

The number of involved glomeruli and the type of injury have prognostic implications and are used to stage kidney involvement (Berden Classification—Table 2). Patients who have >50% sclerotic glomeruli on presentation tend to have the poorest prognosis, while those with only focal involvement (<50% glomeruli involved) have the best prognosis [78,79]. In addition, a scoring system that combines the percentage of normal glomeruli, degree of interstitial fibrosis and tubular atrophy, and the glomerular filtration rate (GFR) at diagnosis can be used to stratify patients into 3 risk groups with prognostic implications [80]. Moreover, patients with MPO-ANCA and MPA have a more pronounced chronic injury manifesting as glomerulosclerosis, interstitial fibrosis, and tubular atrophy when compared to those with PR3-ANCA and GPA [81].

Laboratory workup in patients with AAV typically reveals an increase in systemic markers of inflammation. ANCA testing is routinely performed when there is suspicion for vasculitis. The most common assays for ANCA detection are indirect immunofluorescence (IIF) that utilize alcohol-fixed buffy coat leukocytes or immunoassays such as enzyme-linked immunosorbent assay (ELISA). IIF are not antigen-specific and the use of immunoassays for screening is currently recommended [82]. ANCA subtypes tend to align with the clinical diagnosis, around 75% of patients with GPA have PR3-ANCA; conversely, around 60% of patients with MPA have MPO-ANCA (Table 1). MPO-ANCA predominates in EGPA and increases in frequency in patients with kidney involvement, though patients with EGPA have the highest ANCA negativity rate that reaches >50% [83]. Assays for PR3 and MPO-ANCA carry higher sensitivity and specificity for an AAV diagnosis when compared to IIF [84].

## 4. Management of Kidney Involvement in AAV

The management of patients with ANCA-associated glomerulonephritis (AAGN) typically consists of two treatment phases. An initial “induction” phase that aims to mitigate inflammation and decrease renal scarring. Once disease control is achieved treatment is transitioned to a second “maintenance” phase with a primary goal of preventing disease relapses. The management of AAV depends on disease severity. Severe AAV describes organ- or life-threatening disease, thus patients with AAGN have “severe” AAV by definition. Treatment should be offered to all patients presenting with AAGN regardless of severity of renal injury as a significant proportion of patients recover renal function. For example, in one study, stabilization or improvement in renal function occurred in 68% of 188 patients with a GFR of 20 mL/min or less, and 57% of 96 patients with a GFR of 10 mL/min or less [74]. In addition, even patients who required dialysis at/close to presentation have excellent rates of being liberated from dialysis, ranging from 55 to 90% [74,85,86,87,88]. Patients on dialysis who do not respond after 4 months of induction therapy have a <5% chance of being liberated from dialysis [89]. Thus immunosuppression should be discontinued in those patients unless they suffer from extra-renal disease that warrants continuation of immunosuppression.

### 4.1. Induction of Remission

#### 4.1.1. Glucocorticoids

Despite their many short and long-term adverse effects, glucocorticoids have been a cornerstone in the management of AAGN for decades. A literature review by Hollander et al. published in 1967, described outcomes of 26 patients with GPA that were treated with glucocorticoids and noted improved survival [90]. Currently, most AAGN patients are started on glucocorticoids in the form of oral prednisone (or prednisolone) at a dose of 1 mg/kg/d (maximum 80 mg/d) with a subsequent taper [91]. Patients with rapidly progressive glomerulonephritis or those with alveolar hemorrhage often receive 2–3 “pulses” of intravenous methylprednisolone (IVMP), at a dose of 250–1000 mg per day prior to starting oral prednisone. The optimal dosing and taper strategy for glucocorticoids has not been studied in a randomized controlled trial until recently. The Plasma Exchange and Glucocorticoids in Severe ANCA-Associated Vasculitis (PEXIVAS) study randomized 704 patients in a 2-by-2 factorial design. In addition to comparing the utility of plasma exchange (PLEX) in the management of AAV (discussed further below), it also compared the efficacy of two weight-based regimens of glucocorticoids (standard versus reduced dose) [92] (Table 3). The reduced-dose regimen resulted in a 61% reduction in the cumulative glucocorticoid dose received over the course of 6 months. The reduced-dose regimen was found to be non-inferior to the standard-dose regimen in achieving the study’s primary composite end-point of death or end-stage renal disease (ESRD) at 12 months and to reduce the frequency of severe infections in the first year. Post hoc analysis did not demonstrate any difference in efficacy in patients with severe renal disease, in those with lung hemorrhage, or those treated with rituximab (RTX) versus cyclophosphamide (CYC). The reduced-dose regimen is expected to be adopted as the new standard glucocorticoid regimen in the upcoming 2020 KDIGO (Kidney Disease: Improving Global Outcomes) clinical practice guidelines. Although PEXIVAS did not find any difference in cardiovascular, endocrine or gastrointestinal adverse events between the two glucocorticoid regimens, other studies have demonstrated an increased risk of infections with glucocorticoid use. For example, a retrospective study of 114 patients with severe AAGN (defined as a serum creatinine >5.7 mg/dL or dialysis dependence) that were treated with PLEX, glucocorticoids, and CYC did not demonstrate any improvement in survival, renal recovery, or relapses in patients receiving IVMP compared to patients who did not. However, patients who received IVMP had a higher risk of infection in the first 3 months and a higher risk of diabetes after adjusting for confounders [93]. Other studies found an independent association between glucocorticoids and the risk of major infection [94,95]. Limiting infections is of utmost importance in AAV, since infection—rather than active vasculitis—is the major cause of death in the first year of treatment [96].

#### 4.1.2. Cyclophosphamide

Studies at the National Institute of Health demonstrated that when added to glucocorticoids, CYC resulted in improved rates of remission. CYC revolutionized the management of AAV and improved the prognosis from a fatal disease to one that has >90% remission rate [97,98]. The route of administration of CYC was studied in the pulse versus daily oral cyclophosphamide for induction of remission in ANCA-associated vasculitis (CYCLOPS) trial [99]. CYCLOPS randomized 149 patients with AAGN to prednisolone and either pulse CYC (3 doses of 15 mg/kg given intravenously 2 weeks apart, followed by 15 mg/kg intravenously or orally every 3 weeks until in remission for 3 months) or daily oral CYC (2 mg/kg/d until remission followed by 1.5 mg/kg/d for 3 months). Dose reduction was done if patients were >70 years of age or experienced leukopenia. All patients received maintenance azathioprine (AZA) after completion of induction until 18 months. Patients with a serum creatinine >5.7 mg/dL were excluded from this study. The groups did not differ in the proportion of patients achieving remission by 9 months (88.1% vs. 87.7% in the pulse vs. daily oral groups, respectively). There was also no difference in the time to remission. More patients in the pulse CYC group developed ESRD at 18 months, however the difference did not reach statistical significance (5 vs. 1, *p* = 0.105). There was no difference in the rates of eGFR improvement or death. Patients in the pulse CYC group were more likely to relapse (hazard ratio, HR, 2.01 (confidence interval, CI, 0.77 to 5.30)). Patients receiving pulse CYC had a lower cumulative CYC dose (8.2 vs. 15.9 g) and a lower number of leukopenic episodes. Extended follow-up for a median of 4.3 years demonstrated an increased risk of relapse in the pulse CYC group, but no differences in patient survival and renal function [100].

#### 4.1.3. Rituximab

The use of the anti-CD20 monoclonal antibody rituximab (RTX) for induction of remission in AAV was evaluated in the Rituximab versus Cyclophosphamide for ANCA-Associated Vasculitis (RAVE) trial. RAVE enrolled 197 patients with new or relapsing AAV, 102 of which had kidney involvement. Patients with severe kidney disease (defined as having a serum creatinine >4 mg/dL or dialysis dependence) were excluded. RAVE compared RTX given as 4 weekly infusions at a dose of 375 mg/m^2^ with oral CYC given orally at a dose of 2 mg/kg/d (adjusted for kidney function) followed by AZA at a dose of 2 mg/kg/d. All patients received pulse IVMP followed by a prednisone taper. RTX was non-inferior to CYC in achieving the primary endpoint of complete remission (BVAS (Birmingham Vasculitis Activity Score) = 0 and completion of steroid taper at 6 months) [101]. Interestingly, RTX was more efficacious in patients with relapsing disease (67% vs. 42% patients reaching the primary endpoint in the RTX vs. CYC groups, *p* = 0.01). In patients with renal involvement, RAVE demonstrated similar rates of remission and improvement in GFR between the two groups at 6, 12, and 18 months [102]. In addition, the improvement in eGFR was similar across groups stratified by baseline eGFR [102]. From a serological perspective, the RTX group had more effective B-cell depletion, and a higher rate of ANCA seroconversion (from positive to negative). The higher seroconversion in the RTX group was mainly driven by PR3-ANCA positive patients [101]. Adverse events rates were similar between the two groups. 

RTX and CYC were also compared in a retrospective study of 225 patients with severe renal involvement (defined as eGFR < 30 mL/min/1.73 m^2^) [103]. The RTX and CYC regimens were similar to what was used in RAVE and both treatment groups were treated with glucocorticoids. The two treatment groups had similar rates of remission (BVAS = 0) at 6 months, ESRD or death at 18 months, and ESRD at 24 months. However, RTX had a steroid-sparing effect with more patients in the RTX group achieving complete remission defined as BVAS = 0 AND not on prednisone (17.7% versus 31.7%, *p* = 0.031). Prednisone taper was also faster in the RTX group (median prednisone dose at 6 months, 10.0 versus 2.5 mg, *p* = 0.002; 12 months, 5.0 versus 0.0 mg, *p* = 0.001; 18 months, 3.0 versus 0.0 mg, *p* = 0.036).

#### 4.1.4. Combination CYC and RTX

Combination therapy with CYC and RTX was evaluated in the Rituximab versus Cyclophosphamide in ANCA-Associated Renal Vasculitis (RITUXVAS) trial [104]. Unlike RAVE, patients with severe renal involvement were not excluded in RITUXVAS. Patients participating in RITUXVAS were randomized in a 3:1 ratio to receive combination RTX and IV (intravenous) CYC (RTX: 4 weekly doses of 375 mg/m^2^, IVCYC: 15 mg/kg for 2 doses given with the first and third RTX doses) or IV CYC for 3–6 months followed by AZA. Patients in the combination group did not receive any maintenance immunosuppression. All patients received pulse IVMP and a glucocorticoid taper. There was no difference in the primary endpoint of sustained remission (BVAS = 0 for 6 months) at 12 months. There was no difference in the rate of severe adverse events, infections, or relapses at 12 months. In addition, both treatment groups had a similar improvement in eGFR. 

Several other studies described the use of various combinations of RTX and CYC. A propensity-matched case control study compared patients treated with the CycLowVas regimen, which combined two doses of 1 g of RTX combined with 6 doses of IVCYC (10 mg/kg) to those enrolled in prior European Vasculitis Society (EUVAS) studies, and found a reduced risk of death, progression to ESRD, and relapse [105]. Combination RTX and CYC was also used to limit the use of glucocorticoid. One study that combined RTX and oral CYC and included patients with severe renal disease reported excellent rates of remission with a relatively rapid glucocorticoid-tapering scheduled [106]. In addition, combination therapy was successfully used in prospective cohorts to limit the use of glucocorticoids to 2 weeks or less, even in patients with severe renal disease, while attaining excellent rates of remission (>90%) [107].

#### 4.1.5. Complement

The C5a receptor (C5aR) inhibitor avacopan was tested as a steroid-sparing agent in a phase II randomized controlled trial that enrolled 67 patients with non-severe renal involvement. Patients received induction therapy with CYC or RTX and were randomized into 3 groups; one receiving a standard of care prednisone regimen, a second receiving low dose prednisone, and a third receiving no prednisone. Avacopan was non-inferior to standard-dose prednisone in achieving >50% decrease in BVAS and demonstrated a trend towards faster achievement of remission (BVAS = 0) [108]. Avacopan was further studied in the phase III trial ADVOCATE [73]. ADVOCATE enrolled 331 patients with new or relapsing disease to receive avacopan 30 mg twice daily or prednisone in addition to standard-of-care induction (RTX or CYC). Eighty-one percent of patients had kidney involvement. Avacopan was non-inferior to prednisone in inducing clinical remission at 26 weeks (BVAS = 0 and no glucocorticoids for the preceding 4 weeks). However, avacopan was superior in inducing sustained remission (BVAS = 0 at 26 and 52 weeks, no glucocorticoids in the preceding 4 weeks, no relapses between weeks 26 and 52). In addition, patients receiving avacopan had better improvement in eGFR at 26 and 52 weeks, a decreased rate of glucocorticoid-related adverse effects (as measured by the glucocorticoid toxicity index). The number of serious infections was similar in both groups. 

#### 4.1.6. Plasma Exchange

Evidence for efficacy of plasma exchange (PLEX) in patients with AAGN was demonstrated in the randomized trial of plasma exchange or high-dosage methylprednisolone as adjunctive therapy for severe renal vasculitis (MEPEX) trial [109], which randomized 137 European patients with severe renal disease (serum creatinine >5.7 mg/dL or on dialysis) to receive IVMP or PLEX added to oral CYC and glucocorticoids. MEPEX demonstrated decreased rates of ESRD at 3 and 12 months, with no difference in survival at 12 months. Long-term follow up for a median of 3.95 years did not demonstrate any difference in rates of ESRD or death [110]. The efficacy of PLEX was later assessed in the larger PEXIVAS trial, which randomized 704 patients with a wider eligibility, GFR < 50 mL/min and/or lung hemorrhage, 205 of which had severe renal disease (serum creatinine >5.7 mg/dL or on dialysis) to receive PLEX or no PLEX [92]. There were no significant differences in the primary composite outcome of time to death or ESRD between groups over the average follow-up of 2.9 years. However, a meta-analysis of 10 PLEX randomized controlled studies in AAV that included MEPEX and PEVIAS demonstrated no difference in the rates of the composite of death or ESRD at 3 or 12 months, but showed a decreased overall incidence of ESRD [111]. This meta-analysis did not include the observational study of Casal Moura et al. [103] which did not demonstrate any benefit of PLEX on remission-induction at 6 months, the rate of ESRD and/or death at 18 months, progression to ESRD, and survival at 24 months.

The difference in conclusions from the MEPEX and PEXIVAS studies arises from differences in eligibility, end-points and trial duration. MEPEX enrollment required a kidney biopsy while PEXIVAS did not. However, the majority of PEXIVAS patients were biopsied, but biopsy data is yet to be published. Additionally, PEXIVAS was not powered to detect differences in ESRD as an outcome independent of mortality. A summary of the major trials for induction of remission in AAV can be found in Table 4.

#### 4.1.7. Supportive Management

The use of immunosuppressive medications in management of AAV results in an increase in the rate of infections. Opportunistic infections such as Pneumocystis jirovecii pneumonia have been observed in clinical trials of AAV [112], but unfortunately the use of prophylactic antibiotics has not been consistently reported. The use of prophylactic trimethoprim–sulfamethoxazole was associated with a lower frequency of severe infections in a retrospective series of patients with AAV receiving RTX for induction and should be considered in all patients [113]. In addition, prophylaxis against glucocorticoid-related adverse effects such as osteoporosis, gastric ulcer disease, and candidal infections should be considered in all patients receiving glucocorticoids at high doses or for an extended period of time.

### 4.2. Maintenance of Remission

Patients who achieve remission after receiving induction therapy are typically started on maintenance therapy to decrease the risk of relapse. Until recently, AZA has been the standard of care for maintenance therapy. AZA was compared to oral CYC in the Randomized Trial of Maintenance Therapy for Vasculitis Associated with Antineutrophil Cytoplasmic Autoantibodies (CYCAZAREM) trial, which randomized 144 patients to oral CYC (at a dose of 1.5 mg/kg/d) or AZA (at a dose of 2 mg/kg/d) [114]. The relapse rates were similar between the two groups. AZA was also compared to methotrexate (MTX) in the Azathioprine or methotrexate maintenance for ANCA-associated vasculitis (WEGENT), which demonstrated similar relapse rates and adverse events that persisted with long-term follow up [115,116]. AZA was compared to mycophenolate mofetil (MMF) in the mycophenolate mofetil vs. azathioprine for remission maintenance in antineutrophil cytoplasmic antibody-associated vasculitis: a randomized controlled trial (IMPROVE), which randomized 156 patients to receive AZA or MMF after induction therapy with CYC. IMPROVE showed a decreased risk of relapse with AZA at a median follow up of 39 months (HR 1.69 (95% CI, 1.06–2.70; *p* = 0.03), and a similar severe adverse event rate [117].

More recently, the anti-CD20 monoclonal antibody RTX, has cemented its role as a superior agent for maintenance of remission. The rituximab versus azathioprine for maintenance in ANCA-associated vasculitis (MAINRITSAN) trial randomized 115 patients with mainly new onset disease, who received CYC for induction therapy, to receive AZA (at a starting dose of 2 mg/kg/d for 12 months, with a subsequent taper) or RTX (two initial dose of 500 mg then one dose every 6 months until month 18) [118]. RTX was superior in decreasing the risk of relapse (HR 6.61; 95% CI, 1.56 to 27.96; *p* = 0.002), with a similar severe adverse event rate. Maintenance RTX was compared to AZA in 170 patients with relapsing disease that received induction therapy with RTX in the rituximab as therapy to induce remission after relapse in ANCA-associated vasculitis (RITAZAREM trial). RTX was given at a dose of 1 g every 4 months. RTX was superior in decreasing risk of relapse (HR 0.36, 95% CI 0.23–0.57, *p* < 0.001) [119,120]. 

The optimal dose and duration of RTX maintenance therapy were studied in two additional trials. The comparison of individually tailored versus fixed-schedule rituximab regimen to maintain ANCA-associated vasculitis remission (MAINRITSAN 2) randomized 162 patients to receive a fixed dose of RTX (500 mg every 6 months) or a tailored dosing schedule (one 500 mg dose with >2 fold increase in ANCA titers, or ANCA seroconversion from negative to positive, or peripheral B-cell return). RTX was given until 18 months after randomization. At 28 months, there was no difference in the rate of relapse, or serious adverse events. However, patients in the tailored arm received a lower number of infusions and a lower cumulative RTX dose [121]. Patients completing follow up in MAINRITSAN 2 who were still in remission (*n* = 97) were further randomized to receive 4 additional doses of 500 mg of RTX given every 6 months or placebo in the MAINRITSAN 3 trial. Patients receiving a short RTX treatment were more likely to suffer a relapse (Relapse-free survival HR 7.5, CI 1.67–33.7, *p* = 0.008). In the placebo arm, the majority of relapses occurred in patients with PR3 disease (11/13) [122]. Taken together, these data suggest that low dose rituximab (500 mg) is effective in maintaining remission, and that long-term therapy (for 46 months) is associated with a lower risk of relapse, especially in patients with PR3 ANCA.

Given the excellent outcomes in patients receiving the B-cell depleting RTX, alternative methods for targeting B-cells were studied. In the Efficacy and Safety of Belimumab and Azathioprine for Maintenance of Remission in Antineutrophil Cytoplasmic Antibody–Associated Vasculitis: A Randomized Controlled Study (BREVAS), 105 patients who received induction with CYC or RTX and achieved remission were randomized to receive AZA alone or with add-on belimumab [123]. The study was truncated and did not enroll its initial target of 300 patients mainly due to the change in standard of care maintenance regimen in AAV from AZA to RTX. Belimumab did not reduce the risk of relapse, in part due to an unexpectey low relapse rate in the placebo group. However, despite the small number of events, none of the patients who received induction therapy with RTX suffered a relapse. A summary of the major trials for maintenance of remission in AAV can be found in Table 5.

### 4.3. Patients with End-Stage Renal Disease

Despite the high rate of remission in patients with AAV, the rate of ESRD is around 8% at 6 months and increases to 14% with an average follow up of 7.1 years [124,125]. Patients with MPA are more likely to develop ESRD compared to those with GPA (20.0 vs. 9.0%, respectively, *p* < 0.01). One explanation to this observation is that patients with MPA often have a late presentation with more of them presenting with a eGFR < 50 mL/min (60.0 vs. 36.6%) [124,125]. However, this might not be the sole explanation of their higher risk of ESRD. In another cohort, patients with MPO-ANCA were more likely to develop ESRD even after adjusting for creatinine level at diagnosis (HR 2.64, 95% CI 1.25–5.58, *p* = 0.003) [126]. Once a patient develops ESRD, the risk of renal relapses decreases significantly. In a cohort of patients with AAV, the relapse rate before and after development of ESRD decreased from 0.2 to 0.08 episodes/person-year (*p* = 0.0012). The observed decrease was mainly due to patients with PR3-ANCA (rate decreased from 0.34 to 0.11 episodes/person-year; *p* = 0.0015). Patients with MPO-ANCA had a low relapse rate to start that remained low (pre- and post-ESRD relapse rates of 0.06 and 0.04 episodes/person-year, respectively; *p* = 0.47). Thus early withdrawal of immunosuppression should be considered in patients with ESRD due to ANCA-associated glomerulonephritis, especially those with MPO-ANCA.

Patients with ESRD should be considered for renal transplantation whether or not ANCAs are still present. Patients who are in remission for less than a year before receiving their allograft have a higher mortality (HR 2.3, *p* < 0.05) [127]. After transplantation, the rate of relapse is quite low on current post-transplant immunosuppressive regimens (2.8% per patient-year), however relapse is independently associated with graft loss [128].

### 4.4. Comorbidities and Unmet Needs

Despite the recent advances in controlling disease activity in patients with AAV, little progress has been made in limiting comorbid conditions. An increased risk of infections has been consistently observed in multiple cohorts and clinical trials (reviewed here [112]). The majority of infections tend to involve the respiratory tract, and contribute significantly to mortality [129,130]. Patients with higher disease burden, higher cumulative exposure to glucocorticoids, and those with kidney involvement have the highest risk of infection [112]. An active effort towards minimizing the risk of infection should be pursued and includes minimizing the dose of steroids and prophylaxis against *Pneumocystis jerovicii*. 

Patients with AAV are also at increased risk for venous thromboembolic (VTE) and cardiovascular events as compared to the general population. Around 10% of AAV patients develop VTE events that typically occur in the first year after diagnosis [131]. Kidney, lung, and skin involvement are independent risk factors for VTE events [131]. Similarly, an increased risk of cardiovascular events (coronary artery disease and ischemic stroke) is also observed in patients with AAV [132]. Similar to VTE, most cardiovascular events occur in the first year after diagnosis [132]; both disease activity and ANCA type appear to be associated with cardiovascular risk [133,134]. Thus, aggressive cardiovascular risk factor assessment and modification should be integrated within the therapeutic approach of AAV patients. 

Due to the use of cytotoxic therapy and B-cell depletion in the management of AAV, some patients develop hypogammaglobulinemia. The rate of decrease in immunoglobulin levels is highest after the induction phase of therapy [135]. Patients with a high cumulative CYC dose and those with lower baseline immunoglobulin levels are at higher risk [136]. RTX cumulative dose did not seem to change the risk in one study, however an inherent bias whereby patients with lower IgG levels were less likely to receive repeated RTX may have influenced the results [136]. Additionally, patients in the tailored dose of MAINRITSAN2 had similar changes in immunoglobulin levels compared to those in the fixed-dose arm, despite receiving a lower cumulative dose [121]. However, MAINRITSAN 3 demonstrated that patients treated with RTX for 56 months had higher rates of hypogammaglobulinemia compared to those treated for 28 months [122]. 

Quality of life for people with AAV is an important area of unmet need, impacted by multiple factors. Glucocorticoids are frequently cited as major contributors to impaired quality of life, including both physical and mental wellbeing, while fatigue and pain have been identified by patients as frequent symptoms that are not well addressed by current therapies [137]. The Vasculitis Damage Index (VDI) captures cumulative organ damage through the course of treatment, both from disease activity and adverse effects of therapy, and the Glucocorticoid Toxicity Index is a tool intended to evaluate multiple domains impacted by these medications [138]. However, these instruments do not fully capture patients’ cumulative experience with AAV. Even with well-controlled disease activity, people with AAV have lower quality of life as compared to the general population [139]. Certainly, the hope is that this may improve as more targeted therapies emerge in AAV, but patients’ perspectives are essential to determining this. To better assess patient-reported outcomes (PRO) unique to vasculitis, the Outcome Measures in Rheumatology (OMERACT) Vasculitis Working Group has developed AAV-PRO [140,141,142]. This 29-item tool is now being utilized in clinical trials and practice to better account for patient preferences in treatment decisions and responses. Additionally, The American College of Rheumatology (ACR) and European League Against Rheumatism (EULAR) are currently conducting a Delphi survey of both patients and clinicians to better align patient and physician priorities in evaluating the efficacy and impact of future therapeutic trials.

#### Future Therapies

The success of rituximab in treatment of AAV has paved the way for alternative B-cell directed therapies (Figure 1). Obinutuzumab is a humanized anti-CD20 monoclonal antibody that has superior antibody-dependent cytotoxicity and direct cell death compared to rituximab, and achieves superior peripheral and tissue B-cell depletion [143]. Obinutuzumab has recently demonstrated success in lupus nephritis [144], where autoantibodies play a major role in pathogenesis. B-cell depletion is an attractive treatment strategy in antibody-mediated autoimmune disorders, however, levels of BAFF increase after B-cell depletion, which can preferentially facilitate re-emergence of more autoreactive B-cell populations [145]. Combining B-cell depletion and BAFF inhibition can potentially circumvent this phenomenon. Additionally, combination therapy has the advantage of targeting CD20- plasmablasts, increased depletion of memory B-cells, and improved depletion of B-cell niches (Reviewed elsewhere [145]). B-cell depletion and concomitant BAFF inhibition is being studied in the Rituximab and Belimumab Combination Therapy in PR3 Vasculitis (COMBIVAS, NCT03967925). A similar strategy was also utilized in lupus nephritis and despite not demonstrating improved outcomes, resulted in enhanced the negative selection of autoreactive B cells [146].

Plasma cells do not express CD20 and thus escape depletion by anti-CD20 agents. Patients with AAV have an elevated levels of CD38+ plasma cell that correlate with disease activity [147] and may help predict relapses [148]. Targeting plasma cells in AAV using proteasome inhibitors such as bortezomib has not been studied in clinical trials, largely due to concern about adverse effects. Daratumumab is an anti-CD38 monoclonal antibody that is well tolerated and can potentially be evaluated in patients with AAV.

The interaction between B and T cells is fundamental to the production of class-switched antibodies [149], and thus the B/T cell interaction may also be leveraged therapeutically in AAV. Abatacept is a fusion protein of the Fc region of IgG1 and the extracellular domain of cytotoxic T lymphocyte antigen 4 (CTLA4). It inhibits T cell co-stimulation by binding to CD80/86 on antigen presenting cells, hence blocking its binding to CD28. Abatacept demonstrated efficacy and a steroid-sparing effect in an open-label trial [150]. It is currently being studied in non-severe relapsing GPA (ABROGATE; NCT02108860). Alternatively, the CD40/CD40 ligand interaction of B and T cells also represents a potential therapeutic opportunity. This interaction is being studied in other forms of autoimmune disease such as lupus nephritis, Sjögren’s syndrome, and rheumatoid arthritis [151].

Other potential therapeutic targets in AAV include Bruton’s tyrosine kinase (BTK) and spleen tyrosine kinase (SYK). BTK mediates B-cell receptor signaling and thus affects B-cell growth and maturation [152]. BTK levels are increased in transitional and naïve B-cells in active AAV. In vitro BTK inhibition decreases plasma cell and antibody formation [153]. On the other hand, SYK is a non-receptor tyrosine kinase that mediates signaling by B-cell receptors, Fc receptors, integrins, as well as pattern recognition [154]. In the Wistar Kyoto rat model of AAV, SYK is detected in glomerular lesions of AAV [155]. SYK inhibition decreased in vitro ANCA-induced cellular responses and attenuated the severity of glomerulonephritis [155]. 

Until recently, the innate immune system has not been leveraged therapeutically in AAV. Blocking C5aR demonstrated efficacy in AAV and positioned the complement system as a prime therapeutic target, with Avacopan, as noted above. A monoclonal antibody that targets the C5a molecule, vilobelimab (IFX-1) is being evaluated in patients with AAV (NCT03712345). Finally, the neutrophil is the effector cell in AAV and NETosis is an important mediator of loss of tolerance. Suicidal NETosis (NETosis which involves cell death) is mediated via peptidylarginine deiminase-4 (PAD4). PAD4 inhibition decreased NET formation and MPO-ANCA production in mice [156] and is being developed for clinical use.

## 5. Conclusions

AAV is a multisystem disease that frequently involves the kidneys. Along with genetic and environmental contributions, perturbations in multiple components of the innate and adaptive immune systems have been implicated in its pathogenesis, providing numerous established and potential targets for therapeutic intervention. Despite significant advances in AAV, management remains challenging given the delicate balance between the risks of disease relapse and those of adverse effects from immunosuppression, most notably serious infections. Due to this, the increased use of targeted therapies aimed at decreasing reliance on corticosteroids and other non-specific immunosuppressive agents have moved to the forefront of recent and future trials. This is an exciting time for development of patient-tailored therapeutic approaches. Involving patients in shared decision making to evaluate short- and long-term risks of disease and therapies is essential.

## Figures and Tables

**Figure 1 jcm-10-01446-f001:**
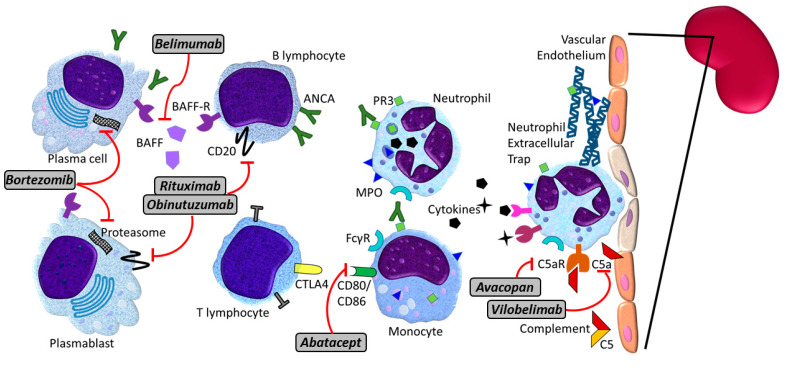
Pathogenesis of ANCA-associated vasculitis and potential targeted therapies. Pathogenesis of ANCA-associated vasculitis is multifactorial, involving numerous immune cells. B lymphocytes feature prominently as producers of ANCAs, with BAFF contributing to maintenance of autoreactive B cells, plasmablasts, and plasma cells. Neutrophils also feature prominently in the pathophysiology, driving much of the tissue damage via injury to the vascular endothelium. Complement factors, particularly C5a, and NETs are also important drivers of vascular inflammation and injury. From a therapeutic perspective, multiple targets have been identified for currently approved therapies and/or future directions, as treatment of ANCA-associated vasculitis becomes more precise and personalized, aiming to minimize the risks of therapy. Among those depicted, rituximab is the only currently approved therapy for ANCA-associated vasculitis, while the remainder represent therapies currently under investigation or potential therapeutic targets. Rituximab and obinutuzumab target CD20 which is expressed in B lymphocytes and plasmablasts. Bortezomib targets the proteasome and primarily target antibody-secreting plasma cells. By targeting BAFF, belimumab impacts the maintenance of autoreactive lymphocytes, plasmablasts, and plasma cells. Abatacept inhibits T-cell stimulation by antigen-presenting cells including B-cells. At the level of the vascular endothelium, vilobelimab and avacopan, target complement factor C5a and C5a receptor, respectively. Abbreviations: ANCA: anti-neutrophil cytoplasmic antibodies. BAFF: B lymphocyte activating factor. C5a: complement factor 5 fragment a. NETs: neutrophil extracellular traps. MPO: myeloperoxidase. PR3: leukocyte proteinase 3.

**Figure 2 jcm-10-01446-f002:**
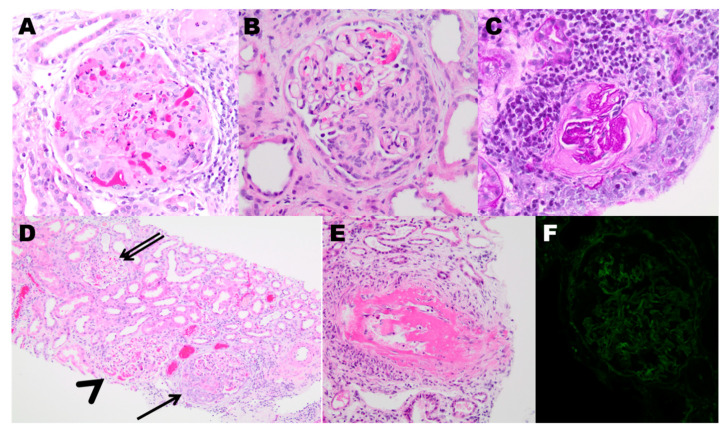
The kidney biopsy in AAV. (**A**) A glomerulus with an active global cellular crescent with segmental necrosis, Hematoxylin and Eosin (H&E), Magnification 200×. (**B**) A glomerulus with segmental fibrocellular crescent, H&E, Magnification 200×. (**C**) A glomerulus with fibrous crescent, Periodic-Acid Schiff (PAS), Magnification 200×. (**D**) Glomeruli in different stages of involvement, including active cellular crescent (single arrow), segmental necrotizing lesion (arrowhead) and uninvolved glomerulus (double arrows), H&E, Magnification 100×. (**E**) Necrotizing vasculitis with severe fibrinoid changes of the arterial wall, H&E, Magnification 200×. (**F**) Immunofluorescence is negative for immunoglobulins.

**Table 1 jcm-10-01446-t001:** Syndromes of AAV.

Disease	Incidence * [7]	ANCA-Positivity	PR3-ANCA	MPO-ANCA	Predominant Organ Involvement	Rate of Renal Involvement [77]	RPGN [77]
GPA	1.9–13	~90%	~75%	~20%	Nose and sinuses, lungs, kidneys, joints, eyes	~70%	~50%
EGPA	0.8–4	~40%	<10%	30–40%	Lungs, upper airways, peripheral nerves, heart, skin	~25%	<15%
MPA	1.5–16	~90%	~25%	~60%	Kidneys	>90%	~65%

* per million person-years. Abbreviations: AAV: ANCA-associated vasculitis. ANCA: Antineutrophil cytoplasmic antibody. PR3: leukocyte proteinase 3. MPO: myeloperoxidase. RPGN: rapidly progressive glomerulonephritis. GPA: granulomatosis with polyangiitis. EGPA: eosinophilic granulomatosis with polyangiitis. MPA: microscopic polyangiitis.

**Table 2 jcm-10-01446-t002:** Classification of ANCA-associated glomerulonephritis *.

Class	Criteria
Focal	≥50% of glomeruli are normal
Crescentic	≥50% of glomeruli have cellular crescents
Sclerotic	≥50% of glomeruli are globally sclerosed
Mixed	Not fulfilling any of the above criteria

* Adapted from [78].

**Table 3 jcm-10-01446-t003:** Glucocorticoid dosing in PEXIVAS (in mg/d).

Week	Standard-Dose Arm			Reduced-Dose Arm		
	<50 kg	50–75 kg	>75 kg	<50 kg	50–75 kg	>75 kg
	pulse	pulse	pulse	pulse	pulse	pulse
1	50	60	75	50	60	75
2	50	60	75	25	30	40
3–4	40	50	60	20	25	30
5–6	30	40	50	15	20	25
7–8	25	30	40	12.5	15	20
9–10	20	25	30	10	12.5	15
11–12	15	20	25	7.5	10	12.5
13–14	12.5	15	20	6	7.5	10
15–16	10	10	15	5	5	7.5
17–18	10	10	15	5	5	7.5
19–20	7.5	7.5	10	5	5	5
21–22	7.5	7.5	7.5	5	5	5
23–52	5	5	5	5	5	5
>52	Investigators’ local practice					

PEXIVAS: The Plasma Exchange and Glucocorticoids in Severe ANCA-Associated Vasculitis study.

**Table 4 jcm-10-01446-t004:** Major clinical trials for induction therapy in ANCA-associated vasculitis.

Trial Name	*N*, Population	Kidney Involvement	Intervention	Control	Primary Endpoint and Conclusion	Other Outcomes
CYCLOPS [99]	149, new AAV	100% (excluded patients with sCr > 5.7 mg/dL)	Pulse CYC (15 mg/kg) every 2–3 weeks	DO CYC (2 mg/kg/d)	No difference in time to remission	Higher risk of relapse in the pulse CYC group on long-term follow up [100]
MEPEX [109]	137, AAGN and sCr > 5.7 mg/dL	100%	PLEX	IV MP	Renal recovery at 3 months. PLEX superior	No difference in long-term outcomes [110]
PEXIVAS [92]	704, new or relapsing	98% (29% with sCr > 5.7 mg/dL)	(1) PLEX * (2) low-dose GC *	(1) no PLEX * (2) standard-dose GC *	(1) Death or ESRD at 12 months. No difference (2) No difference in efficacy	No difference in subgroup analysis for ESRD, death, alveolar hemorrhage
RAVE [101]	197, new or relapsing	52%, (excluded patients with sCr > 4 mg/dL)	RTX (375 mg/m^2^) for 4 weekly doses	PO CYC (2 mg/kg/d) followed by AZA	Remission (BVAS = 0) and completion of steroid taper at 6 months. No difference	RTX better for relapsing disease
RITUXVAS [104]	44, new AAGN	100%	RTX (375 mg/m^2^) for 4 weekly doses + IV CYC (15 mg/kg) for 2 doses	IV CYC (15 mg/kg/d) for 6–10 doses followed by AZA	Sustained remission (BVAS = 0 for 6 months). No difference	
ADVOCATE [73]	331, new or relapsing	81%	Avacopan with RTX or CYC	Prednisone with RTX or CYC	(1) Clinical remission (BVAS = 0 and no steroids ^#^ at week 26). No difference (2) Sustained remission (BVAS = 0 at weeks 26 and 52 and no steorids ^#^ at week 52). Avacopan superior	

* 2 × 2 factorial design. ^#^ no steroids for 4 weeks prior to endpoint. Abbreviations: AAV: ANCA-associated vasculitis. AAGN: ANCA-associated glomerulonephritis. AZA: azathioprine. sCr: serum creatinine. IV: intravenous. DO: daily oral. CYC: cyclophosphamide. ESRD: end-stage renal disease. PLEX: plasma exchange. MP: methylprednisolone. RTX: Rituximab. GC: glucocorticoids. BVAS: Birmingham Vasculitis Activity Score.

**Table 5 jcm-10-01446-t005:** Major clinical trials for maintenance therapy in ANCA-associated vasculitis.

Trial Name	*N*, Population	Kidney Involvement	Induction Agent	Intervention	Control	Primary Endpoint and Conclusion	Other Outcomes
CYCAZAREM [114]	155, new AAV	94% (excluded those with sCr > 5.7 mg/dL)	DO CYC	AZA (2 mg/kg/d)	DO CYC (1.5 mg/kg/d)	Relapse at 18 months. No difference	MPA relapses less frequently than GPA
MAINRITSAN [118]	115, new or relapsing	70%	Pulse CYC	RTX (500 mg) days 0, 14 then every 6 months	AZA (2 mg/kg/d) for 1 year followed by a taper	Major organ relapse at 28 months. RTX superior.	Most relapses occurred after B-cell return
MAINRITSAN 2 [121]	162, new or relapsing	72%	CYC or RTX or MTX **	Tailored * RTX (500 mg)	Scheduled RTX (500 mg) every 6 months	Relapse at 28 months. No difference	No difference in adverse events. Fewer RTX doses in tailored arm
MAINRITSAN 3 [122]	97 patients from MAINRITSAN 2	63%	CYC or RTX or MTX **	RTX (500 mg) every 6 months for 18 months	Placebo	Relapse-free survival at 28 months. Less relapses with extended treatment	Most relapses occurred in GPA patients
RITAZAREM [120]	170, relapsed AAV	N/A	RTX	RTX (1 g) every 4 months for 20 months	AZA (2 mg/kg/d)	Time to relapse. RTX superior.	
IMPROVE [117]	156, new AAV	N/A	CYC	AZA (2 mg/kg/d) for 1 year followed by a taper	MMF (2 g/d) for 1 year followed by a taper	Relapse-free survival. Relapses more common with MMF	
BREVAS [123]	105, new or relapsing	N/A	CYC or RTX	Belimumab (10 mg/kg) + AZA	Placebo + AZA	Time to protocol-specified event ^‡^. No difference.	No relapses in patients induced with RTX and receiving belimumab

* RTX re-dosed with B-cell return, or ANCA seroconversion (negative to positive), or increase in ANCA ** only one patient induced with MTX ^‡^ Birmingham Vasculitis Activity Score (BVAS) of ≥6, or presence of ≥1 major BVAS item, or receipt of prohibited medications for any reason, resulting in treatment failure Abbreviations: AAV: ANCA-associated vasculitis. D: day. DO: daily oral. CYC: cyclophosphamide. AZA: Azathioprine. RTX: Rituximab. MTX: Methotrexate. MMF: mycophenolate mofetil. N/A: data unavailable.

## Data Availability

Not applicable.

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
