# Peer review of "ANCA-Associated Vasculitis: An Update"

_jcm, 2021, doi:10.3390/jcm10071446_

Round 1
Reviewer 1 Report
The manuscript, “ANCA- Associated Vasculitis: An Update,” by Almaani et. al. provides a comprehensive review of current and emerging therapies in ANCA vasculitis. The manuscript is thorough, expertly constructed, and easy to read. I only have a few minor suggestions:
P6, line 214-215. Would clarify that immunosuppressive therapy should be discontinued after 3 months if patients remain on dialysis and do not have extrarenal disease that warrants immunosuppression.
P6, line 239-240. Can the authors add a table comparing the PEXIVAS standard and reduced dose glucocorticoid regimens?
P12, Comorbidities and unmet needs
Please add discussion regarding long term use of B cell depleting agents and risk of hypogammaglobulinema.
Author Response
Reviewer 1,
Thank you for taking the time to review our manuscript and for your valuable input. Please find a point-by-point response to your comments below.
- Comment 1: “P6, line 214-215. Would clarify that immunosuppressive therapy should be discontinued after 3 months if patients remain on dialysis and do not have extrarenal disease that warrants immunosuppression.”
- Response: we added a comment to clarify this point (P7, line 217)
- Comment 2: “P6, line 239-240. Can the authors add a table comparing the PEXIVAS standard and reduced dose glucocorticoid regimens?”
- Response: we added the PEXIVAS steroid dose protocol in Table 3.
- Comment 3: “P12, Comorbidities and unmet needs. Please add discussion regarding long term use of B cell depleting agents and risk of hypogammaglobulinema.”
- Response: we added a paragraph discussing hypogammaglobulinemia (P14 lines 502-511)
- Additional changes: we added a discussion of the ADVOCATE trial to ensure the review is up to date.
An updated version of the manuscript is attached.
Reviewer 2 Report
The review by Almaani et al gives an excellent update and overview on ANCA vasculitis. The manuscript is very well written and I have only few comments:
Major:
Lines 110-118: I advise to also mention the potential role of Th17 cells in AAV
Lines 179-186: please briefly discuss histological differences between PR3-ANCA and MPO-ANCA; the treatment in the future may eventually become different between these two serotypes, especially once we get hold of therapeutic options concerning the development of renal fibrosis
Line 201, part 3: why is the title directed towards renal AAV? Most cited studies included patients with and without renal disease.
Lines 457-476: please also mention quality of life studies, since AAV has serious impact on this
Minor:
Line 132: please rephrase the last sentence
Part 3.4.1: please refer to figure 1 in the text
Author Response
Reviewer 2,
Thank you for taking the time to review our manuscript and for your valuable input. Please find a point-by-point response to your comments below.
- Comment 1: “Lines 110-118: I advise to also mention the potential role of Th17 cells in AAV.”
- Response: we added a comment about the potential role of Th17 cells in early and late glomerular injury (P4, lines 114-116)
- Comment 2: “Lines 179-186: please briefly discuss histological differences between PR3-ANCA and MPO-ANCA; the treatment in the future may eventually become different between these two serotypes, especially once we get hold of therapeutic options concerning the development of renal fibrosis.”
- Response: we added a statement about the more pronounced features of chronic injury in patients with MPO-AAV (P6, lines 188-190).
- Comment 3: “Line 201, part 3: why is the title directed towards renal AAV? Most cited studies included patients with and without renal disease.”
- Response: We agree with the reviewer that the trials included patients with renal and extra-renal disease. However in this review, we chose to focus on kidney involvement and did not discuss extra-renal involvement (sinopulmonary disease, management of non-severe disease).
- Comment 4: “Lines 457-476: please also mention quality of life studies, since AAV has serious impact on this”
- Response: we added a paragraph about quality of life studies (P14-15, lines 512-530)
- Comment 5: “Line 132: please rephrase the last sentence.”
- Response: The phrase was taken out and full description of the ADVOCATE study was added to the section discussing induction therapy (P10, lines 343-353)
- Comment 6: “Part 3.4.1: please refer to figure 1 in the text.”
- Response: We added it on P15 line 533
An updated version of the manuscript is attached.
Regards
Salem Almaani on behalf of the authors
Reviewer 3 Report
This is a very thorough and well-written review on ANCA vasculitis. The authors do an outstanding job of summarizing the complex pathophysiology. Figure 1 is extremely effective in showing pathogenesis as well as therapeutic targets. They thoroughly reviewed the literature on management including induction and maintenance therapies and summarized the key findings in well-organized table format. I have no major suggestions. The authors may want to include data on infection prophylaxis during treatment as well as specific populations such as overlap ANCA/anti-GBM disease and pediatric patients.
Author Response
Reviewer 3,
Thank you for taking the time to review our manuscript and for your valuable input. Please find a point-by-point response to your comments below.
- Comment 1: “The authors may want to include data on infection prophylaxis during treatment.”
- Response: we added a paragraph discussing antibiotic prophylaxis (P11 lines 385-395)
- Comment 2: “include specific populations such as overlap ANCA/anti-GBM disease and pediatric patients”
- Response: We agree with the reviewer that there are specific populations that require alteration of therapeutic approaches. Those include patients with concomitant anti-GBM disease, lupus nephritis, membranous nephropathy, and IgA nephropathy. We think that discussion of those populations deserves a thorough review that is beyond the scope of this manuscript.
An updated version of the manuscript is attached.
Regards
Salem Almaani on behalf of the authors